# An Electrochemical Sensor for Sulfadiazine Determination Based on a Copper Nanoparticles/Molecularly Imprinted Overoxidized Polypyrrole Composite

**DOI:** 10.3390/s23031270

**Published:** 2023-01-22

**Authors:** Manahil Babiker Elamin, Shazalia Mahmoud Ahmed Ali, Houda Essousi, Amani Chrouda, Laila M. Alhaidari, Nicole Jaffrezic-Renault, Houcine Barhoumi

**Affiliations:** 1Department of Chemistry, Faculty of Science Al-Zulfi, Majmaah University, Majmaah 11952, Saudi Arabia; 2Laboratory of Interfaces and Advanced Materials, Faculty of Sciences, Monastir University, Monastir 5000, Tunisia; 3Institute of Analytical Sciences, University of Lyon, UMR CNRS 5280, 69100 Villeurbanne, France

**Keywords:** sulfadiazine, copper nanoparticles, molecularly imprinted polymer, overoxidized polypyrrole, differential pulse voltammetry

## Abstract

To protect consumers from risks related to overexposure to sulfadiazine, total residues of this antibacterial agent in animal-origin foodstuffs not exceed international regulations. To this end, a new electrochemical sensor based on a molecularly imprinted polymer nanocomposite using overoxidized polypyrrole and copper nanoparticles for the detection of sulfadiazine is elaborated. After optimization of the preparation of the electrochemical sensors, their differential pulse voltammetric signal exhibits an excellent stability and reproducibility at 1.05 V, with a large linear range between 10^−9^ and 10^−5^ mol L^−1^ and a low detection limit of 3.1 × 10^−10^ mol L^−1^. The produced sulfadiazine sensor was successfully tested in real milk samples. The combination of the properties of the electrical conduction of copper nanoparticles with the properties of the preconcentration of the molecularly imprinted overoxidized polypyrrole allows for the highly sensitive detection of sulfadiazine, even in real milk samples. This strategy is new and leads to the lowest detection limit yet achieved, compared to those of the previously published sulfadiazine electrochemical sensors.

## 1. Introduction

Sulfadiazine (4-amino-N-2-pyrimidinylbenzenesulfonamide, SDZ) is an antibacterial agent widely used in human and veterinary medicines to treat infectious diseases such as ear infections, urinary tract diseases, toxoplasmosis, pneumonia, and malaria [1,2]. The over exposure to SFX can cause physiological problems such as liver damage, antibiotic resistance, and gene damage. Thus, to protect consumers from risks related to SDZ residues, the European Union Council Regulations set a safe limit stating that the combined total residues of SDZ must not exceed 100 ng/g in animal-origin foodstuffs [3]. Following international regulation, the maximum allowable concentration of SDZ in milk is 0.07 ppm [4]. Therefore, the determination of trace level of SDZ in milk and other animal products used for human consumption has become an important and urgent task. Previously, various techniques have been used for SDZ detection, such as high-performance liquid chromatography, ultraviolet [5,6], UV-Visible [7,8], and fluorescence [9] spectroscopies, and other hyphenated techniques [10,11,12]. The principal limitations of these techniques are their high costs, the requirement of a long preparation time, and the necessity of sophisticated technical skills. However, electrochemical sensors based on molecular imprinting polymer (MIP) have gained more attention in recent years, due to their simple manipulation, high selectivity, and portability [13,14,15,16]. MIP is formed by the polymerization or electropolymerization of functional monomer in the presence of the target analyte (template molecule), followed by the extraction of the template molecule from the polymer matrix via washing steps, leaving specific cavities which are complementary to the template molecule [17,18,19,20]. MIP has been widely applied in pharmaceutical analysis [21,22], food analysis [23], and environmental analysis [24]. A molecularly imprinted polyacrylate-based sensor for SDZ detection was developed by Liu et al. [25]. The determination of SDZ through differential pulse voltammetry was obtained in the range of 4 to 50 × 10^−6^ M, with a detection limit of 0.68 × 10^−6^ M. The method was successfully applied to determine sulfadiazine in milk and milk powder. A carbon paste electrode combined with polyacrylate MIP was used for the determination of sulfadiazine in milk and human serum samples by Sadeghi et al. [26]; the obtained dynamic range was from 2.0 × 10^−7^ M to 1.0 × 1 0^−4^ M, and the detection limit was 1.4 × 10^−7^ M.

Conductive polymer polypyrrole (PPy) is widely employed in chemical sensors and biosensors [27,28]. PPy has many advantages, such as being easy to prepare through electropolymerization, having good redox properties, being environmentally stable, and exhibiting good conductivity and good biocompatibility [20]. PPy can be overoxidized in a sodium hydroxide (NaOH) solution during several scans of cyclic voltammetry. This process plays a positive role during the formation of MIP since it introduces oxygen groups (carboxyl, carbonyl, and hydroxyl) into the OPPy backbone [29,30], which could be suitable for the recognition of sulfadiazine positive groups. However, this process makes the polymer film less conductive. To compensate for this drawback, we propose the subsequent electrodeposition of copper nanoparticles (CuNPs) that will help to enhance the film conductivity. CuNPs exhibit good conductivity, high chemical and mechanical stability, and a high specific surface area that makes them suitable for electrochemical sensors. Some researchers have proven that copper nanoparticles electrochemically deposited on the MIP surface are useful in the sensitive detection of drugs and pollutants [31,32,33].

Many electrochemical sensors were designed for the detection of sulfadiazine [25,26,34,35,36,37]. Some sensors were based on direct electrochemical detection on different types of electrode materials: glassy carbon [38], boron-doped diamond [39], or bismuth film [38]. Different nanomaterials were used for enhancing the sensitivity of the detection of SDZ: rare earth compounds [36,40,41], phthalocyanines [42,43], manganese and zinc oxides [44,45,46,47], vanadium and bismuth sulfides [38,41], metallic nanoparticles [40], and carbonaceous nanomaterials [38,39,42,44,48,49,50,51]. In addition to the various MIP-based sensors designed, different polymers were also used for MIP fabrication: polyacrylate [25,26], PEDOT [34], and PPy [35]. PPy was overoxidized in an NaOH solution, and a graphene oxide@covalent organic framework (GO@COF) nanocomposite was added for signal amplification of the MIP film. In this work, the unique conductive properties of CuNPs were combined for the first time with the molecularly imprinted OPPy film for sulfadiazine electrochemical determination. The analytical performance of the optimized CuNPs/MIP-OPPy/GCE sensor was evaluated by differential pulse voltammetry and validated for its detection in real spiked milk samples.

## 2. Materials and Methods

### 2.1. Chemical Reagents

Pyrrole was purchased from Merck, Germany, and then purified through distillation. Copper sulfate, sodium nitrate, sulfadiazine, acetone, sulfuric acid, hydrochloric acid, dapsone, sulfamethoxazol, dopamine, ascorbic acid, and uric acid were obtained from Sigma Aldrich, Germany. Hexacyanoferrate (II/III) was obtained from FlukaChemika.

### 2.2. Instrumentation

The samples were characterized by Fourier transform infrared spectroscopy (FT-IR) using a Perkin Elmer 1600 FTIR spectrometer (Perkin Elmer Waltham, MA, USA) and scanned over the wave number range between 400 and 4000 cm^−1^.

The surface morphology of the modified electrodes was characterized by the use of a ZEISS EVO MA 25 (Carl Zeiss S.A.S. Rueil-Malmaison, France) scanning electron microscope (SEM). HR-TEM images were obtained using a JEM-2100 analytical electron microscope (JEOL Europe S.A.S., Croissy-sur-Seine, France) operating at an accelerating voltage of 200 kV. For HR-TEM images, the CuNPs/MIP-OPPy composite was transferred from the electrode surface to the TEM grid.

Electrochemical measurements were carried out using an Autolab electrochemical analyzer, model PGSTAT 302N (Eco Chemie, Utrecht, The Netherlands). A three-electrode system was employed, consisting of a glassy carbon working electrode (GCE, diameter 3 mm), an Ag/AgCl/KCl_(3M)_ electrode used as a reference electrode, and a platinum electrode used as a counter electrode. Electrochemical experiments were controlled by using Nova version 1.5 software. Cyclic voltammetry was carried out at a scan rate of 100 mV/s. For the detection of SDZ in 0.1 M H_2_SO_4_ solution, the parameters for DPV (differential pulse voltammetry) were: increment—5 mV; amplitude—50 mV; sample period—100 ms; pulse time—10 ms.

### 2.3. Preparation of the SDZ Imprinted Polypyrrole Film

Before use, the GCE was cleaned according to the procedure presented in [52]. The modification process consisted of the electropolymerization of 0.1 M pyrrole monomer in the presence of sulfadiazine in a 0.1 M H_2_SO_4_ solution, using cyclic voltammetry at a scan rate of 100 mV/s, in the potential range from −0.2 to +1.4 V for 10 cycles. After electropolymerization, the sensor was rinsed with water and dried at room temperature. The recognition sites in the imprinted matrix were formed after removing the template from the imprinted film. For this purpose, the film was overoxidized by cyclic voltammetry (CV) in a 0.1 M NaOH solution, until current stabilization was reached, leaving imprints in the polymer film corresponding to the template shape. This process made the surface of the electrode less conductive. The loss of PPy electroactivity was compensated for by incorporating CuNPs into the film through cyclic voltammetry. CV curves of MIP-OPPy/GCE in the presence of CuSO_4_ were obtained, under different scanning cycles (from 1 to 30), in a 0.1 M Na_2_SO_4_ solution containing 10^−3^ M CuSO_4_. After the electrodeposition of the CuNPs on the modified electrode, it was rinsed with deionized water and dried. The procedure for the fabrication of CuNPs/MIP-OPPy/GCE is schematically presented in Figure 1. Respectively, the non-imprinted polymer membrane (NIP) was also fabricated in the same way, but without SDZ.

### 2.4. Determination of Sulfadiazine in Real Samples

Reduced-fat milk samples were purchased from Zulfi market. Appropriate quantities of SDZ were spiked in these samples. A total of 25 mL of milk were centrifuged at 2000 rpm for 30 min to precipitate fat. A total of 10 mL of acetonitrile was added to the supernatant for its deproteinization. The mixture was then centrifuged at 4000 rpm for 15 min, after which the supernatant was then mixed with 25 mL of H_2_SO_4_ solution (pH 3) [53].

## 3. Results and Discussion

### 3.1. Elaboration of CuNPs/MIP-OPPy Sensor and NIP Polypyrrole Film

#### 3.1.1. Electropolymerization of MIP and NIP Polypyrrole Film

The cyclic voltammograms obtained during the electropolymerization of the MIP polypyrrole film (in the presence of SDZ) and of the NIP polypyrrole film (in the absence of SDZ) are presented in Figure 1. It appears that the exchanged charge quantity is lower for MIP compared to that of NIP.

The SEM image of bare GCE (Figure 2A) is used as the control for comparison. The surface of NIP/GCE (Figure 2B) shows a uniform formation of a polypyrrole film. From Figure 2C, the thickness of MIP in the presence of SDZ was less important than that of NIP under identical conditions, in agreement with the exchanged charge quantity.

#### 3.1.2. Template Extraction through Overoxidation of MIP Polypyrrole Film

Using organic reagents as the eluent for the template removal requires abundant organic reagents and is also time-consuming. In addition, it is very difficult to completely remove the template in a short time. The elimination of the cationic charge in the polymer backbone by further oxidation in high pH conditions (overoxidation) allows the template to be removed from the overoxidized PPy film, and this procedure does not require any organic reagents. CV is used for the overoxidation of the PPy film in the 0.1 M NaOH solution; the cyclic voltammogram is shown in Figure 3. Comparing the first cycle to the fourth, the peaks at −0.01 V and 0.85 V disappeared in the fourth cycle, which is related to the oxidation process of PPy. At the same time, the anodic peak ascribed to SDZ oxidation at 1.1 V, through the reaction presented in Figure 2, also disappeared. After SDZ elution from the PPy film, through the oxidation reaction, a molecularly imprinted film is fabricated. The 6th and 7th cyclic voltammograms are almost identical, which proves that PPy is completely oxidized. After overoxidation, a porous structure appears on the surface of MIP-OPPy, as shown in the SEM image (Figure 2D).

#### 3.1.3. Electrodeposition of Copper Nanoparticles

The cyclic voltammograms obtained during the electroreduction of copper are presented in Figure 4. The reduction peak of copper occurs at −0.25 V. This CV curve is similar to that obtained for the electrodeposition of CuNPs on pectin [54]. In this work, the authors obtained crystalline copper with (111), (200) and (220) diffraction peaks. The EDX spectra of the composite film is presented in Figure 5A, showing that copper is a major compound of this composite. The obtained copper nanoparticles were observed using HR-TEM, as presented in Figure 5B. Their diameter is between 3 and 4 nm.

### 3.2. FTIR Characterization

Figure 6 shows the FTIR spectra of the SDZ powder, PPy/SDZ, and MIP-OPPy matrices. The absorption peaks observed at 3351 and 3422 cm^−1^ for sulfadiazine powder are attributed to the bending vibration of primary NH_2_ symmetric and asymmetric stretching, respectively [55,56]. The peaks at 1327 and 1581 cm^−1^ are assigned to C=N and -CNS of the SDZ molecules, respectively (Figure 6a, Table 1) [57]. The peaks observed at 666 and 785 cm^−1^ are assigned to the asymmetric stretching vibration of the SO_2_ group [58]. Before the removal of the sulfadiazine molecule, the FTIR spectrum of PPy/SDZ (Figure 6b, Table 1) shows two weak adsorption peaks at 672 and 785 cm^−1^, which are attributed to the asymmetric vibration of SO_2_-N in the SDZ molecules [59]. It indicates that the template SDZ molecule was successfully entrapped in the polymer film through hydrogen bonds. The absorption bands of the template molecules disappeared in the MIP-OPPy spectrum, which can be attributed to the ejection of SDZ from the film (Figure 6c, Table 1). Thus, the FTIR characterization has been efficiently adapted to confirm that the SDZ molecules were successfully entrapped and removed from the polymer structure. On the other hand, the FTIR spectra of PPy/SDZ and MIP-OPPy possess the same absorption bands, differing only in intensity. The absorption peaks at 2910 and 2980 cm^−1^ are attributed to the symmetric and asymmetric N-H stretching of the amine group of the pyrrole group. The band at 1750 cm^−1^ is attributed to the symmetric stretching vibration of C-H.

### 3.3. Effect of the Experimental Conditions for SDZ Detection

To maximize the sensitivity of the CuNPs/MIP-OPPy/GCE sensor for the determination of SDZ, some experimental parameters needed to be optimized, including the number of electropolymerization scan cycles, the concentrations of sulfadiazine and the monomer, incubation time, and pH value. The DPV signal of SDZ in a 0.1 M H_2_SO_4_ solution (pH = 2) containing 10^−6^ mol L^−1^ SDZ was reported as the response of the MIP sensor to the different parameters.

#### 3.3.1. Effect of Py Monomer and SDZ Template Concentrations

The Py monomer concentration during the electrodeposition process can influence the electrochemical response of the MIP sensor [60]. Different films were grown from solutions containing a constant concentration of SDZ (10^−3^ M) and with varied concentrations of the pyrrole monomer, between 1 mM and 200 mM. Figure 7A reports the variation in the MIP sensor response for the different pyrrole concentrations. The response increases when increasing the pyrrole concentration up to 0.1 mol L^−1^ and for higher concentrations, the response decreases. We can deduce that the optimum monomer concentration is about 0.1 mol L^−1^.

The effect of the SDZ concentration in the electropolymerization solution on the MIP sensor response was further examined. The concentrations of SDZ were between 10^−4^ mol L^−1^ and 10^−2^ mol L^−1^. As shown in Figure 7B, the MIP sensor response increases with the increase in the SDZ concentration between 10^−4^ mol L^−1^ and 10^−3^ mol L^−1^. When the template concentration was higher than 10^−3^ mol L^−1^, the current decreased. According to this result, the optimum template concentration was chosen to be 10^−3^ mol L^−1^.

#### 3.3.2. Effect of the Number of Electropolymerization Cycles

Beyond the concentrations of the monomer and the template, another important parameter that may affect the electroanalytical performance is the MIP film thickness, controlled by the number of scan cycles. Figure 8 shows the effect of the number of scan cycles on the MIP sensor response. It can be seen that the current response first increases, reaching a maximum at 15 cycles, then decreases for a larger number of cycles. A total of 15 cycles were then selected to elaborate the SDZ MIP sensor.

#### 3.3.3. Effect of the Incubation Time

The incubation process is an important parameter that influences the sensitivity of the MIP sensor [61]. Incubation times of the MIP sensor varied from 5 to 25 min. Figure 9 shows that the MIP sensor response increases gradually with the increasing incubation time, and when the incubation time is higher than 15 min, the MIP sensor response remains almost stable, suggesting that the SDZ adsorption equilibrium is reached. The optimal incubation time was then selected as 15 min.

#### 3.3.4. Effect of the Scan Rate

The effect of the scan rate on the oxidation peak current of SDZ was investigated by cyclic voltammetry, in the presence of 10^−6^ mol L^−1^ SDZ in 0.1 mol L^−1^ H_2_SO_4_ (pH = 2), with scan rates ranging from 20 to 200 mV/s (Figure 10). The oxidation peak currents increased with an increase in the scan rate, while their oxidation peak potentials gradually shifted to positive values. A plot of I_pa_ versus the square root of the scan rate (*v*^1/2^) in the range of 20 to 200 mV/s yielded a straight line, with a good correlation coefficient (R^2^ = 0.99235). This relationship transduces a diffusion-controlled oxidation reaction.

#### 3.3.5. Effect of the pH Value

Figure 11A shows the effect of pH on the response of the MIP sensor within the pH range of 2.0–7.0. The potential of the oxidation peak is shifted in the positive direction when the pH increases, which is in agreement with the oxidation reaction presented in Figure 2. The current decreases when pH levels range from pH 2.0 to higher pH values (Figure 11B). Based on these results, a pH equal to 2.0 was chosen for the detection of SDZ. The best affinity of SDZ for the MIP-OPPy film is obtained in solutions with a pH value of 2, showing that the main interactions are hydrogen bonds [60].

### 3.4. Analytical Performance of the MIP Sensor

Figure 12A shows the differential pulse voltammetry (DPV) curves recorded for different SDZ concentrations in 0.1 mol L^−1^ H_2_SO_4_ (pH = 2). The DPV curves are recorded in the range of 10^−9^ to 10^−5^ mol L^−1^. It can be seen that the peak current increased with increasing SDZ concentrations. Figure 12B shows the calibration curve as the peak current versus the -log ([SDZ]) recorded for the MIP sensor and the NIP sensor. The detection limit LOD of the MIP sensor is 3.1 × 10^−10^ mol L^−1^, according to the formula 3σ/S, σ being the background of the blank, and S being the sensitivity. The linear relationship has been observed in the concentration range from 10^−9^ to 10^−5^ mol L^−1^, with a correlation coefficient of R^2^ = 0.98307 and R^2^ = 0.98574 for NIP and MIP sensors, respectively. A high analytical response was obtained with a sensitivity of 3.78 μA mol L^−1^ for the MIP electrode, in comparison with the NIP electrode, with a sensitivity of 0.77 μA mol L^−1^. The imprinting factor of the MIP sensor is 4.9. The sensitivity of MIP without CuNPs was 1.387 μA mol L^−1^. The integration of CuNPs into the MIP sensor enhances the conductivity of the MIP film, which in turn can dramatically improve the sensitivity of the imprinted sensor. Compared with previously published works on electrochemical SDZ sensors, the CuNPs/MIP-OPPy/GCE sensor designed in this study presents the lowest detection limit and the largest dynamic range (Table 2).

Four sensors were prepared independently, on the same day and under the same optimal conditions, to detect 10^−6^ mol L^−1^ SDZ in the H_2_SO_4_ solution (pH 2). The relative standard deviations (RSD, n = 4) of the DPV measurements was 2.1%, which indicates the excellent reproducibility of the CuNPs/MIP-OPPy/GCE-based sensor. The stability of the developed sensor was also tested. The DPV response of the MIP sensor was measured after an interval of five days. As shown in Figure 13, the MIP sensor exhibits only a slight decrease (2.4%) in the DPV signal after 25 days, demonstrating high stability and good repeatability.

The response of the MIP sensor and of the NIP sensor for different interfering substances of SDZ was measured. The MIP and NIP sensors were introduced into individual solutions of sulfadiazine (10^−6^ mol L^−1^, pH 2) and of some analog substances, such as sulfamethoxazol (SFD), dapsone (DDS), dopamine (DA), uric acid (UA), and ascorbic acid (AA) at 10^−6^ mol L^−1^ concentration, and the DPV signal was measured. The responses are reported in Figure 14. The SDZ DPV signal of the MIP sensor was 1.4 times higher than that of the SFD, and 3 times higher than that of the DDS, these molecules being analogs of SDZ; the specificity of the imprints in OPPy film is demonstrated. Nevertheless, due to its similarity of structure, SFD is partly recognized by the imprints in the MIP-OPPy film. DA, AU and AA are interfering substances in biological samples. The DPV signals of the MIP and NIP sensors are quite similar, showing that they present non-specific adsorption on both films.

### 3.5. Real Sample Analysis

Real sample analysis was performed by the MIP sensor in SDZ-spiked milk samples. The final concentrations in the spiked samples, after pretreatment, were first controlled using HPLC/MS/MS. The recoveries of the SDZ and RSD values for three determinations are summarized in Table 3. The unexpected results of an 83% recovery rate for one spiked milk sample can be explained by a fouling of the MIP surface by remaining proteins, which could hinder the adsorption of sulfadiazine on the MIP surface; thus, the concentration of sulfadiazine was underestimated. For two concentrations, the obtained recoveries indicate that the MIP sensor can be used successfully for the determination of SDZ in milk samples.

## 4. Conclusions

A novel and efficient electrochemical sensor based on a CuNPs/MIP-OPPy nanocomposite for SDZ detection has been designed. The performance of this sensor was evaluated by conducting a study of its electrochemical response to sulfadiazine in aqueous samples using differential pulse voltammetry. The MIP sensor offers a linear current response to SDZ in the range of 10^−9^ to 10^−5^ mol L^−1^, with a low limit of detection of 3.1 × 10^−10^ mol L^−1^. The results indicate that the incorporation of copper nanoparticles provides better sensitivity (3.78 μA mol L^−1^) for the MIP electrode, the sensitivity of the MIP sensor without CuNPs being 1.387 μA mol L^−1^. The MIP electrode exhibits good reproducibility, repeatability, and stability for more than 25 days for the determination of SDZ, and it was applied for SDZ quantification in spiked real milk samples.

## Data Availability

Not applicable.

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
