# Peer review of "An Electrochemical Sensor for Sulfadiazine Determination Based on a Copper Nanoparticles/Molecularly Imprinted Overoxidized Polypyrrole Composite"

_sensors, 2023, doi:10.3390/s23031270_

Round 1

Reviewer 1 Report

Babiker Elamin et al. prepared copper nanoparticles/molecularly imprinted overoxidized polypyrrole composite-based sensor for sulfadiazine determination in milk samples.  However, the manuscript needs major revision for publication consideration. My specific comments are detailed below:

1. The title of the manuscript is very long and unclear. Please change to:

Copper Nanoparticles/Overoxidized Polypyrrole Composite as a Sensor for Sulfadiazine Determination in Milk Samples.

2. Avoid abbreviations in the abstract.

3. The abstract should include the following points: a summary of your findings; new concepts and innovations demonstrated; a brief restatement of your hypotheses; a comparison with literature-reported results, and possible future work.

4. Table 1 must be transferred to the results discussion

5. Did you use pyrrole without purification ? 

6. In the " Preparation of the SDZ imprinted polypyrrole film" section? Why did the use of  pyrrole monomer (0.1 M)?

7. Please provide the EDX mapping of composite for confirmation of preparation of composite.

8. Please assign the characteristic peaks in the FTIR spectra.

Author Response

We would like to address special thanks to the reviewer for insightful comments on the original version of our paper. We strongly believe that these comments lead to a real improvement of the present work.

Babiker Elamin et al. prepared copper nanoparticles/molecularly imprinted overoxidized polypyrrole composite-based sensor for sulfadiazine determination in milk samples.  However, the manuscript needs major revision for publication consideration. My specific comments are detailed below:

  1. The title of the manuscript is very long and unclear. Please change to:

Copper Nanoparticles/Overoxidized Polypyrrole Composite as a Sensor for Sulfadiazine Determination in Milk Samples.

The title of the manuscript was modified as follows : « An electrochemical Sensor for Sulfadiazine Determination based on Copper Nanoparticles/Molecularly Imprinted Overoxidized Polypyrrole Composite »

  1. Avoid abbreviations in the abstract.

Abbreviations in the abstract were canceled.

  1. The abstract should include the following points: a summary of your findings; new concepts and innovations demonstrated; a brief restatement of your hypotheses; a comparison with literature-reported results, and possible future work.

The abstract was reformulated, as follows :

Abstract: To protect consumers from risks related to overexposure to sulfadiazine, total residues of this antibacterial agent in animal-origin foodstuffs should follow international regulation. To this end, a new electrochemical sensor based on molecularly imprinted polymer nanocomposite using overoxidized polypyrrole and copper nanoparticles for the detection of sulfadiazine is elaborated. After optimization of the preparation of the electrochemical sensors, their differential pulse voltammetric signal exhibits an excellent stability and reproducibility at 1.05 V, with a large linear range, between 10-9 and 10-5 mol L-1, and a low detection limit of 3.1x10-10 mol L-1. The produced sulfadiazine sensor was successfully tested in real milk samples. The combination of the properties of electrical conduction of copper nanoparticles with the properties of preconcentration of the molecularly imprinted overoxidized polypyrrole allows the highly sensitive detection of sulfadiazine, even in real milk samples. This strategy is new and leads to the lowest detection limit, compared to the previously published sulfadiazine electrochemical sensors.

  1. Table 1 must be transferred to the results discussion

Table 1 was transfered to the Results and Discussion part.

  1. Did you use pyrrole without purification ? 

Pyrrole was purified through distillation. This point was added in line 92.

  1. In the " Preparation of the SDZ imprinted polypyrrole film" section? Why did the use of  pyrrole monomer (0.1 M)?

In §3.3.1. Effect of Py monomer, it is shown that the concentration of 0.1 M pyrrole monomer is optimum.

  1. Please provide the EDX mapping of composite for confirmation of preparation of composite.

EDX spectra of the composite film is presented in Figure 5A, showing that copper is a major compound of this composite. HR-TEM mapping of CuNPs on the composite surface is presented in Figure 5B.

  1. Please assign the characteristic peaks in the FTIR spectra.

Characteristic peaks were assigned on the FTIR spectra (Figure 6).

Reviewer 2 Report

In this work, the authors established Copper Nanoparticles/Molecularly Imprinted Overoxidized Polypyrrole Composite-Based Sensor for Sulfadiazine Determination in Milk Samples. Frankly speaking, the manuscript is lack of novelty and some data should be checked carefully. But the analytical performance seems competitive, such as lower LOD compared with previous works. Some important issues should be addressed before considering for publication. I recommend major revision of manuscript.

Specific comments:

1.      The novelty of this work should be illustrated further in the introduction. Why did the authors conduct this experiment?

2.      Last paragraph of the Introduction should be rewritten.

3.      As a research paper, the number of references is too much. The list of previous reports is not necessary.

4.      Generally, the nanomaterials were modified before the MIP preparation. In this work, CuNPs were deposited after the removal of the template. How much is the thickness of the nanolayer modification? Does it affect the MIP sites? The SEM image of CuNPs/MIP/GCE needs to be added.

5.      CuNPs were prepared using cyclic voltammetry. How to conduct the HR-TEM image of CuNPs?

6.      “25 mL of milk were centrifuged at 2000 rpm for 30 min to precipitate fat. Then the milk was treated by using 10 mL acetonitrile to deproteinize it. Subsequently, the mixture was placed in a centrifuge at 4000 rpm for 15 min. The supernatant was removed and transferred to a solution of 25 mL of H2SO4 solution (pH 3).” What is the amount of supernatant. What is the spiking concentration. After the sample pretreatment, what is the concentration of sample for the sensing detection?

7.      The descriptions of Fig. 2D and 3E were not found in the text.

8.      Comparing the first cycle to the fourth one, the peaks at -0.01 V and 0.85 V disappeared in the fourth cycle, which is related to the oxidation process of polypyrrole. Why are these two peaks not present on the MIP preparation curve?

9.      The error bars should be added in these Figures throughout the manuscript.

10.  Check the unit of adding amount in table 3.

11.  Please explain the unexpected results of 83% recovery rate.

12.  The sensing results should be validated by the commonly used method.

Author Response

We would like to address special thanks to the reviewer for insightful comments on the original version of our paper. We strongly believe that these comments lead to a real improvement of the present work.

In this work, the authors established Copper Nanoparticles/Molecularly Imprinted Overoxidized Polypyrrole Composite-Based Sensor for Sulfadiazine Determination in Milk Samples. Frankly speaking, the manuscript is lack of novelty and some data should be checked carefully. But the analytical performance seems competitive, such as lower LOD compared with previous works. Some important issues should be addressed before considering for publication. I recommend major revision of manuscript.

Specific comments:

  1. The novelty of this work should be illustrated further in the introduction. Why did the authors conduct this experiment?

In the last paragraph of the introduction, we have shown that our approach is original (lines 85-86):

In this work, the unique conductive properties of CuNPs were combined for the first time with the molecularly imprinted OPPy film for sulfadiazine electrochemical determination.

  1. Last paragraph of the Introduction should be rewritten.

The last paragraph of the introduction was rewritten (lines 75-89)

  1. As a research paper, the number of references is too much. The list of previous reports is not necessary.

We focused the reference list on electrochemical sensors for the detection of sulfadiazine. For comparison of the analytical performance, it is necessary to cite the published articles.

  1. Generally, the nanomaterials were modified before the MIP preparation. In this work, CuNPs were deposited after the removal of the template. How much is the thickness of the nanolayer modification? Does it affect the MIP sites? The SEM image of CuNPs/MIP/GCE needs to be added.

EDX spectra of the composite film is presented in Figure 5A, showing that copper is a major compound of this composite. HR-TEM mapping of CuNPs on the composite surface is presented in Figure 5B.

  1. CuNPs were prepared using cyclic voltammetry. How to conduct the HR-TEM image of CuNPs?

For HR-TEM images, the CuNPs/MIP-OPPy composite was transferred from the electrode surface to the TEM grid. This point was added in the experimental part (lines 105-106).

  1. “25 mL of milk were centrifuged at 2000 rpm for 30 min to precipitate fat. Then the milk was treated by using 10 mL acetonitrile to deproteinize it. Subsequently, the mixture was placed in a centrifuge at 4000 rpm for 15 min. The supernatant was removed and transferred to a solution of 25 mL of H2SO4 solution (pH 3).” What is the amount of supernatant. What is the spiking concentration. After the sample pretreatment, what is the concentration of sample for the sensing detection?

The final concentrations in spiked samples after pretreatment were firstly controlled using HPLC/MS/MS. This point was added in the text (lines 408-409)

  1. The descriptions of Fig. 2D and 3E were not found in the text.

Figure 2D is cited in 205. Figure 2E is cancelled and replaces by Figure 5B.

  1. Comparing the first cycle to the fourth one, the peaks at -0.01 V and 0.85 V disappeared in the fourth cycle, which is related to the oxidation process of polypyrrole. Why are these two peaks not present on the MIP preparation curve?

      The MIP preparation curve (Figure 1) is obtained in 0.1M H2SO4 solution whereas the overoxidation of polypyrrole is obtained in 0.1 M NaOH. The electrochemical reactions are quite different.

  1. The error bars should be added in these Figures throughout the manuscript.

Error bars were added to all the figures

  1. Check the unit of adding amount in table 3.

The adding amount was transduced in final concentration

  1. Please explain the unexpected results of 83% recovery rate.

The unexpected results of 83% recovery rate for this spiked milk sample can be explained by a fouling of the MIP surface by remaining proteins that could hinder the adsorption of sulfadiazine on MIP surface, the concentration of sulfadiazine was underestimated. This point was added in the text (lines 410-412)

  1. The sensing results should be validated by the commonly used method.

The final concentrations in spiked samples after pretreatment were firstly controlled using HPLC/MS/MS. This point was added in the text (lines 408-409).

Reviewer 3 Report

1. Relationship between crystal structure and sensing performance should be well discussed in this paper.

2. Some work on the sensing of Sulfadiazine (4-amino-N-2-pyrimidinylbenzenesulfonamide, SDZ) has been updated and cited, such as CrystEngComm, 2022, 24, 7157–7165 and CrystEngComm, 2021, 23, 8043–8052; “The absorption peaks observed at 3351 and 3422 cm-1 for sulfadiazine powder are attributed to the bending vibration of primary NH2 symmetric and asymmetric stretching, respectively. This should be cited related refs, such as Inorganics, 10(2022) 202 and Mesopor. Mat, 341(2022) 112098.

3. I suggest the author give a more detail explanation in the introduction “… a clearer storyline should be introduced by choosing a suitable entry point to explain the reasons for designing these material.”

4. Some grammar and spelling errors in the text should be avoided.

5. The main peaks of functional groups should be marked clearly in Fig5.

6. Please also give a Table for comparison on the sensing LOD on such documents and this work.

Author Response

We would like to address special thanks to the reviewer for insightful comments on the original version of our paper. We strongly believe that these comments lead to a real improvement of the present work.

  1. Relationship between crystal structure and sensing performance should be well discussed in this paper.

The CV curve obtained for the electrodeposition of copper nanoparticles is similar to that presented in Ref 57. In this work, the authors obtained crystalline copper with (111), (2 0 0) and (2 2 0) diffraction peaks.

  1. Some work on the sensing of Sulfadiazine (4-amino-N-2-pyrimidinylbenzenesulfonamide, SDZ) has been updated and cited, such as CrystEngComm, 2022, 24, 7157–7165 and CrystEngComm, 2021, 23, 8043–8052; “The absorption peaks observed at 3351 and 3422 cm-1 for sulfadiazine powder are attributed to the bending vibration of primary NH2 symmetric and asymmetric stretching, respectively. This should be cited related refs, such as Inorganics, 10(2022) 202 and Mesopor. Mat, 341(2022) 112098.

We focused the reference list on electrochemical sensors for the detection of sulfadiazine, it is why the suggested references related to MOF-based fluorescence sensors were not cited.

Inorganics, 10(2022) 202 was cited as Reference 57

  1. I suggest the author give a more detail explanation in the introduction “… a clearer storyline should be introduced by choosing a suitable entry point to explain the reasons for designing these material.”

In the last paragraph of the introduction, we have shown that our approach is original (lines 85-86):

In this work, the unique conductive properties of CuNPs were combined for the first time with the molecularly imprinted OPPy film for sulfadiazine electrochemical determination.

  1. Some grammar and spelling errors in the text should be avoided.

Grammar and spelling errors were corrected.

  1. The main peaks of functional groups should be marked clearly in Fig5.

Characteristic peaks were assigned on the FTIR spectra (Figure 6).

  1. Please also give a Table for comparison on the sensing LOD on such documents and this work.

Table for comparison of LOD of published electrochemical sensors with our sensor is presented in the results and discussion part as Table 2.

Round 2

Reviewer 1 Report

It can be published in current form.

Reviewer 2 Report

My concerns have been well addressed by the authors.

Reviewer 3 Report

accept